# Bioelectrical, Anthropometric, and Hematological Analysis to Assess Body Fluids and Muscle Changes in Elite Cyclists during the Giro d’Italia

**DOI:** 10.3390/biology12030450

**Published:** 2023-03-15

**Authors:** Álex Cebrián-Ponce, Alfredo Irurtia, Jorge Castizo-Olier, Manuel Vicente Garnacho-Castaño, Javier Espasa-Labrador, Zeasseska Noriega, Marta Carrasco-Marginet

**Affiliations:** 1INEFC-Barcelona Sports Sciences Research Group, Institut Nacional d’Educació Física de Catalunya (INEFC), Universitat de Barcelona (UB), 08038 Barcelona, Spain; acebrian@gencat.cat (Á.C.-P.); airurtia@gencat.cat (A.I.); franciscojavierespasa@gencat.cat (J.E.-L.); znoriega@gencat.cat (Z.N.); 2School of Health Sciences, TecnoCampus, Pompeu Fabra University, 08302 Barcelona, Spain; jcastizo@tecnocampus.cat; 3DAFNiS Research Group (Pain, Physical Activity, Nutrition and Health), Campus Docent Sant Joan de Déu, University of Barcelona, 08830 Sant Boi de Llobregat, Spain; mavigarcas@gmail.com; 4Faculty of Health Sciences, Valencian International University (VIU), 46002 Valencia, Spain

**Keywords:** BIVA, cycling, phase angle, sport, plasma volume

## Abstract

**Simple Summary:**

Giro d’Italia is one of the most demanding races in cycling tour, in which, during the 3 weeks of competition, several physiological adaptations occur. This study aimed to analyze these changes with different methods (bioelectrical, anthropometrical, and hematological analysis) in order to see how body fluids vary in the whole body as the competition progresses, and also in three specific muscle groups (quadriceps, hamstrings, and calves). There were three checkpoint assessments: at the beginning, middle, and end of the race. Results indicated that bioelectrical impedance vector analysis is sensitive to the body fluid changes. A proper homeostatic adaptation was detected during the first half of competition, but the long-term, high intensity exercise would result in profound fluid loss and distribution in the soft tissues. Furthermore, most of these changes happened in the extracellular water compartment, indicating that the state of the cell membranes was maintained in good condition. The calves are the muscle group most sensitive to this analysis. In conclusion, bioelectrical impedance vector analysis is a reliable, non-invasive, and practical method to assess physiological adaptations in athletes.

**Abstract:**

This study aimed to characterize and monitor the body fluid and muscle changes during the Giro d’Italia in nine elite cyclists via bioelectrical (whole-body and muscle-localized) anthropometric and hematological analysis. There were three checkpoint assessments: at the beginning, middle, and end of the race. The Wilcoxon signed-rank test was used to compare the data at baseline and follow up. The Spearman correlation was used to explore relationships between variables. Hotelling’s T^2^ test was used to determine bioelectrical differences in the complex vector. Bodyweight did not change during the competition, despite bioelectrical and hematological data indicating that at the first half of the race, there was a fluid gain, and in the second half a fluid loss occurred, reaching baseline values. These changes were especially prevalent in the extracellular water compartment. Significant correlations between whole-body bioelectrical vector changes and red blood cell parameter changes were reported. The muscle group most sensitive to changes were the calves. Quadriceps, hamstrings, and calves reported a PhA decrease trend during the first half of the race, and an increase during the second half. Bioelectrical impedance vector analysis appears to be sensitive enough to detect hydration and cellular integrity adaptions induced by competitions as demanding as the Giro d’Italia.

## 1. Introduction

In professional cycling, “Tour de France”, “Giro d’Italia”, and “Vuelta a España” are the most prestigious and demanding multistage races included in the International Cycling Union (UCI) World Tour. In these events, the cyclists ride a long distance of ~3500–4500 km over 3 weeks in 21–23 daily stages of different types. 

In the current literature, there are different reports available showing the physiological changes produced by the high demands of these races [1,2,3,4]. However, knowledge is still sparse about the extent of potential dehydration due to prolonged strenuous cycling and its hematological acute effects [5]. The difficulty of systematically intervening with invasive tests in a professional group of cyclists during the 3 weeks of hard and strenuous competition could be one of the main reasons for the lack of data availability.

In this context, bioelectrical impedance vector analysis (BIVA) emerges as a method to assess body hydration and cellular status [6,7]. The classic BIVA approach works with the same basis of bioelectrical impedance analysis (BIA), but instead of using predictive equations to obtain quantitative data, it relies on the analysis of raw bioelectrical values (resistance, R; reactance, Xc) standardized by conductor length. From these parameters, the derived impedance (Z) and the phase angle (PhA) can also be obtained, representing the vector length and vector direction, respectively [8]. According to classic BIVA, Z is inversely related to total body water (TBW) [6], whereas PhA is considered an indicator of cellular health and cell membrane integrity and is inversely related to the extracellular/intracellular water (ECW/ICW) ratio [9]. All interpretations should be based on the interpretation of Z and PhA jointly, along with the vector position on the Resistance-Reactance (RXc) graphs [10]. BIVA can be performed through different protocols, depending on the device and the distribution of the electrodes [6]: whole-body BIVA (WB-BIVA) may be used to assess the composition of the entire body, whereas muscle-localized BIVA (ML-BIVA) may be used to assess the composition or status of one specific muscle or portion of muscle. In the sports field, some research has performed using both WB-BIVA and ML-BIVA to assess hydration changes and the muscle impact after a training session or competition of different sports [6,8,11].

In some studies made with elite cyclists during the Grand Tour competitions, a vector length shortening without changes in the direction has been reported, indicating a body fluid gain with the extracellular/intracellular water ratio remaining constant at the end of the race [2,3]. Another study made with segmental BIVA, which is an alternative method analyzing body limbs separately, reported that the body compartment most stressed by cycling races was the legs, showing a significant decrease in the PhA [12] and, therefore, an increase in the ECW/ICW ratio at the thighs.

Despite the advantages that BIVA presents as a method to assess hydration, there are other methods available, such as the haematological and serum parameter analysis, which has greater precision and accuracy than BIVA, but which is also invasive, costly, and demanding in terms of time. Regarding this method, Corsetti et al. [13] analysed some haematological parameters during the Giro d’Italia 2011 in order to define the variations across the stage race. They found that the hemoglobin (Hb) and the hematocrit (Hct) decreased during the first half of the race (indicating body fluid gain) with a stabilization in the second half due to an hemodilution typical of endurance sports. Therefore, some similarities can be observed between the interpretations of the hematological changes in relation to the bioelectrical changes reported in cycling grand events. This relationship could better identify changes in body fluid distribution and composition associated with adaptations after exercise and sport. Monitoring hydration status is thereby important for better sport performance, especially in endurance sports [14].

This study aimed to analyze body fluid and muscle changes evoked by the Giro d’Italia stage race by using bioelectrical (whole-body and muscle-localized) anthropometrical and hematological assessments. We hypothesized that both WB-BIVA and ML-BIVA are sensitive enough to detect subtle chronic fluid variations that occur in the body due to this extremely long and demanding race. We also compared the BIVA characteristics of these elite professional cyclists with a healthy adult male Italian reference population [15].

## 2. Materials and Methods

### 2.1. Subjects

This prospective, quasi-experimental study involved 9 professional cyclists (age: 27.9 ± 2.4 years; height (H): 181.4 ± 6.1 cm; body mass (BM): 70.5 ± 6.1 kg; body mass index (BMI): 21.4 ± 0.4 kg/m^2^) from the Cannondale Pro-Cycling Team who participated in the 2013 Giro d’Italia, featured in Table 1. They were all in good health, as assessed by their team physician. No drugs or supplements influencing fluid balance were taken by athletes; only non-steroidal anti-inflammatory drugs and antibiotics were administered when needed, according to the “no needle” policy; and, lastly, therapies and drugs were allowed only for evident illnesses. Official urine and blood controls were performed by anti-doping agencies before the start of the race (day −3) and during the final phase (day 20), and one subject had an additional check on day 5. The study was approved by the Ethics Committee for Clinical Sport Research of Catalonia (Ethical Approval Code: 006/CEICGC/2023) and written informed consent was obtained from each cyclist prior to their participation. All the procedures were in accordance with the Declaration of Helsinki [16].

### 2.2. Experimental Design

The multistage race took place from 4 to 26 May 2013. The route’s main features are reported in Table 1. The mean completion time for the 9 participants was 86:47:29 h (range, 84:12:10–88:15:51 h) with a mean speed of 35.7 km/h. 

As shown in Figure 1, bioelectrical, anthropometric, and hematological measurements were taken a day before the start (PRE), on the first resting day (MID), and on the final day (POST). All measurements were performed between 7am and 9am in a fasted condition. There were 2 rest days (days 10 and 16) plus the cancelled stage of day 21.

### 2.3. Bioelectrical Assessment

R and Xc were measured using a phase-sensitive multifrequency impedance plethysmograph (DS Medica, Human-Im PLUS Impedance meter unit 80C32, Milano, Italy) that emits an alternating sinusoidal electric current of 1 mA at 5 operating frequencies within the range of 5.5, 10, 50, 100, and 300 kHz, and PhA at 10, 50, and 100 kHz, continuously autocalibrated every 5 min with a known impedance provided by the manufacturer. The 50 kHz frequency was selected because of its best signal to noise ratio [7]. Bioelectrical variables were obtained by the same trained examiners in a thermo-neutral room (25 °C; ±40 relative humidity), as recorded by a portable weather station (Kestrel Weather K4500, Nielsen-Kellerman, Boothwyn, Pennsylvania, PA, USA).

Subjects were tested with their arms and legs kept from touching the body by non-conductor foam objects to prevent adduction or the crossing of the limbs. Bioelectrical measurements were recorded after a stabilization period of 5 min, in which the cyclists remained lying motionless to ensure the proper distribution of body fluids. Three measurements were performed every 60 s and the average value was used for the final calculations. Before placing the electrodes, the skin was prepared by shaving the area to remove hair, and rubbing the area with gel and cleaning it with alcohol.

To analyze WB-BIVA, injector electrodes were placed on the dorsal surface of the right hand (proximal to the third metacarpal-phalangeal joint) and foot (proximal to the third metatarsal-phalangeal joint). The detector electrodes were placed proximally 5 cm from the injector ones in order to prevent interaction between electric fields, which could lead to an overestimation of the impedance values.

ML-BIVA was performed in both limbs. Once the symmetry between limbs was verified, the mean value of all parameters was calculated for each muscle analyzed. In the quadriceps, injector electrodes were placed at 5 cm from the anterior superior iliac spine and from the superior pole of the patella, and the detector electrodes were spaced 5 cm from the injector ones, proximal to the center of the muscle. In the hamstrings, injector electrodes were placed 5 cm from the ischial tuberosity and from the popliteal line, and the detector electrodes were spaced 5 cm from the injector ones, proximal to the center of the muscle. In calves, injector electrodes were placed 5 cm from the popliteal line and from the intermalleolar line, and the detector electrodes were spaced 5 cm from the injector ones, proximal to the center of the muscle. 

Z was calculated as √(R^2^ + Xc^2^), and PhA as tan^−1^ (Xc/R · 180°/π). R, Xc and Z were adjusted by height (R/H, Xc/H, Z/H). TBW, ICW and ECW were estimated by the athlete-specific prediction models proposed by Matias et al. [17] on the basis of height, BM, R, Xc and sex. RXc Z-score graph was used to plot the cyclists on the three checkpoints (PRE, MID, POST) regarding the 50%, 75%, and 95% tolerance ellipses for the general Italian healthy adult male population [15]. In accordance with the classic BIVA, the reference population selected may be divided into 4 quadrants: upper left quadrant representing athletes, lower left quadrant representing obese individuals, upper right quadrant representing lean individuals, and lower right quadrant representing cachectic individuals. Furthermore, RXc paired graphs were used to compare bioelectrical differences over time and inter-limb symmetry. In these graphs, ellipses overlapping the origin indicate no differences, whereas non-centered ellipses indicate significant changes.

### 2.4. Anthropometric Assessment

The anthropometric measurements were taken according to the standard criteria of the International Society for the Advancement of Kinanthropometry (ISAK) [18]. The following anthropometric material was used: a telescopic measuring rod (Seca 220^®^, Birmingham, UK, measuring range: 85–200 cm; accuracy: 1 mm) to measure the height; a scale (Seca 710^®^; Birmingham, UK; previously calibrated capacity: 200 kg; accuracy: 50 g) to measure BM; and an anthropometric tape (Lufkin Executive^®^, Lufkin, TX, USA, accuracy 1 mm) to determine the position where electrodes (Red DotTM, 3M Corporate Headquarters, St. Paul, MN, USA) were placed. A waterproof pen was used to mark the anatomical sites for electrodes. BMI was calculated as BM/H^2^ (kg/m^2^).

### 2.5. Hematological Measurements

The UCI and World Anti-Doping Agency (WADA) rules for sample collection and transportation of specimens were followed. Evacuated tubes (BD Vacutainer Systems, Becton-Dickinson, Franklin Lakes, NJ, USA) were used for hematological tests (BD K_2_ EDTA 3.5 mL tubes), while 7 mL plain tubes (BD SSTII Advance) were used for clinical chemistry tests. Immediately after drawing, tubes were inverted 10 times and stored in a sealed box at 4 °C. Controlled temperature was assured during transportation using a specific tag (Libero Ti1, Elpro, Buchs, Switzerland) for temperature measurement and recording. The K_2_ EDTA-anticoagulated blood was homogenized for 15 min prior to being analyzed. Plain tubes were centrifuged for 10 min, 1300× *g*, at 4 °C and the sera was stored at −80 °C until analysis. Clinical chemistry analyses were performed in a single batch and by the same technician. Hematological tests, which included Hb concentration and Hct, were performed on a Sysmex XE 2100 (Sysmex, Kobe, Japan). During the study, the analyzers were regularly calibrated and controlled by both internal and external quality control schemes. The imprecision of hematological tests was <2%. A day-by-day control of imprecision was performed on the Sysmex instrument by using fresh blood during the study, giving an imprecision of <1.6%. Estimated plasma volume (EPV) was calculated on the basis of the BM and Hct, using the Kaplan formula [19]. Biochemical analyses of sodium (Na^+^) and potassium (K^+^) (indirect potentiometry) were performed in an Advia 2400 automatic device (Siemens Medical Solutions Diagnostics, Tarrytown, NY, USA), following the International Federation of Clinical Chemistry’s Committee on Reference Systems for Enzymes [20]. The osmolality was measured by an osmometer performed in an Osmo Station OM-6050 Arkray from Menarini (based on the decrease in cryoscopic point).

### 2.6. Statistical Analysis

Descriptive data are presented as mean ± standard deviation. After testing each variable for the normality of the distribution (Shapiro–Wilks test), differences in all variables over time (PRE, MID, and POST) were tested using the non-parametric Friedman test and, in the case of differences, the Wilcoxon signed-rank test. The magnitude of changes was computed as delta percent values (∆%). Spearman’s rank coefficient of correlation was applied to determine possible associations between hematological and bioelectrical delta percent changes. An RXc Z-score graph was used to characterize the sample. RXc paired graphs and paired one-sample Hotelling’s T^2^ test were used to identify bioelectrical changes over time and to check the bioelectrical inter-limb symmetry. The significance level was set at *p* < 0.05. SPSS (Chicago, IL, USA, ver. 21) and BIVA software [21] was used for data analysis.

## 3. Results

The BIVA point graph (Figure 2) indicates that almost all cyclists were between the 50 and 75 percentiles of the tolerance ellipses of the male adult Italian reference population during the three checkpoints. Its vector locations were also shifted to the left and upwards, and, therefore, with a greater PhA and body cell mass, corresponding with the athletic quadrant.

Several bioelectrical, anthropometric, and hematological changes were reported during and after the race (Table 2). Body fluid changes were not detected by BM changes (*p* > 0.4) nor by the water prediction equations (*p* > 0.064). Whole-body BIVA indicated a non-significant fluid gain during the first half of the race (PRE-MID) by means of a Z/H increase (−1.1 ± 3.8%, *p* = 0.515), and a significant body fluid loss in the second half of the race (MID-POST) as Z/H considerably increased (5.5 ± 6.0%, *p* = 0.008). In the entire race (PRE-POST), cyclists experienced a significant Z/H increase (4.2 ± 4.6%, *p* = 0.038). The predictive ECW/ICW ratio remained constant throughout the race (*p* > 0.083), even though some PhA changes were observed. Regarding the hematological analysis, fluid gain in PRE-MID is significantly indicated by the decrease of Hb and Hct (*p* = 0.008) and the increase of the EPV (*p* = 0.011) and Na^+^ (*p* = 0.027). In MID-POST, the increase of Hb (*p* = 0.007) and Hct (*p* = 0.015), as well as the decrease of EPV (*p* = 0.05), indicate a fluid loss. Osmolality progressively increased over the race (9.3 ± 10.8%; *p* = 0.011), indicating dehydration at the end of the race.

Whole-body BIVA paired graph (Figure 3A) detected a non-significant trend of fluid gain by a shortening of the vector in PRE-MID (T^2^ = 8.3, *p* = 0.1), and a fluid loss in MID-POST by a vector lengthening (T^2^ = 10.9, *p* < 0.001). The resultant PRE-POST vector change was a significant lengthening (T^2^ = 15.6, *p* < 0.001). As seen in Figure 3B–D, the most sensitive muscle group to vector changes were the calves, with a significant shortening in PRE-MID (T^2^ = 60.4, *p* < 0.001) and a lengthening in MID-POST (T^2^ = 11, *p* < 0.001). The resultant vector change was a significant shortening (T^2^ = 12.1, *p* < 0.001). Quadriceps and hamstrings did not report significant vector migrations. All three muscle groups reported a similar PhA trend, decreasing in PRE-MID (2.5–3.1%) and increasing in MID-POST (4.3–7.7%).

Correlation analysis showed associations between some hydration markers changes with some WB-BIVA parameters changes during the race. Hb changes were correlated with Xc/H changes at PRE-MID (r = 0.782, *p* = 0.013) and with R/H, Xc/h, Z/h (r = 0.711, *p* = 0.032; r = 0.833, *p* = 0.005; r = 0.711, *p* = 0.032; respectively) at MID-POST. Hct (r = 0.733, *p* = 0.025) and EPV (r = 0.733, *p* = 0.025) changes were also correlated with Xc/H changes only at MID-POST.

## 4. Discussion

The present study highlighted several body changes evoked by the demands of the race, indicating a body fluid gain during the first half and a bigger posterior fluid loss during the second half. These changes were especially seen in the whole-body bioelectrical and hematological analysis that correlate between them, especially with the red blood cell parameters (Hb and Hct) and EPV. BM assessment and predictive equations could not detect hydration changes in the competition.

To our best knowledge, this is the first study to apply ML-BIVA in a group of elite cyclists. Results indicated that the muscle group most sensitive to the hydration changes produced by the race were the calves, with the same bioelectrical trend as the rest of the body.

Results confirmed the hypothesis that both WB-BIVA and ML-BIVA are sensitive to detecting fluid variations, encouraging BIVA as a method to monitor physiological adaptations in sport.

### 4.1. Athlete’s Characterization

A greater PhA of our subjects in comparation with the reference group is reported in Figure 2, reflecting a better cell function [22] and differing fluid distribution [23], as has already been amply reported in athletes from different sports [24,25,26,27]. These data agree with the specific body composition of elite athletes (greater ICW/ECW ratio) characterized by a better soft tissue hydration, better cellular membrane exchange capacity, and better physical working capacity than the healthy adult population [28]. In fact, TBW, ICW, and PhA could be good indicators for muscle performance in all age groups [29,30]. Changes in these parameters, but especially in the intracellular water compartment, may alter the power generation capacity [31]. 

### 4.2. Whole-Body Changes

Our results did not show a significant decrease in BM both during and at the end of the race, confirming that the energetic balance between energy intake and energy expenditure was achieved. Current evidence establishes that, in the context of athletic performance, it is normal to lose >2% of BM, and this does not cause a clinically relevant reduction in TBW [5]. In fact, it is reported via urine specific gravity analysis that most cyclists of the highest competition level are not well hydrated prior to and also during an important competition [32].

For a better monitoring of hydration, then, it is convenient to use other methods beyond the BM analysis. Only one study analyzed quantitatively TBW in cycling grand tours (by means of BIVA), and reported a non-significant decrease (0.4 ± 1.1%, *p* > 0.05), but also a possible ICW reduction due to the reduction of body cell mass and PhA [33]. In neither the just mentioned study nor the present one (PRE-MID: 1.2 ± 2.1%, *p* = 0.075; PRE-POST: 1.0 ± 1.5%, *p* = 0.086), there seem to be magnitudes of change in TBW at a level sufficient to be considered relevant to the detriment of performance [5]. Likewise, it is possible to experience no BM or TBW changes, and yet to observe significant changes in the different water compartments [5]. Therefore, it is necessary to apply other methods to assess hydration [34]. Analyzing body water compartments (Table 2) through the athlete-specific prediction models [17], cyclists maintained hydration balance, and the homeostatic stability remained constant until the end of the race, despite the fact that there are non-significant trends. 

However, both raw bioelectrical values and whole-body vector migration did report changes over the race: in the first half, there was a shortening trend of the vector, and in the second half there was significant lengthening, above baseline value. These results are similar to other cycling studies [2,3], and they indicate that although proper homeostatic adaptation was detected during the first week of competition, the long-term, high-intensity exercise would have resulted in TBW changes and its distribution in soft tissues, affecting R and Z. Moreover, Xc may be compromised by the oxidative stress associated with the cell damage and the antioxidant depletion generated by this type of physical exercise [35]. With respect to PhA, it describes a similar pattern to the one shown in Pollastri et al. [2], decreasing in the early stages of the race and subsequently increasing until reaching baseline values. PhA changes could reflect water compartment changes not detected by the predictive equations, indicating that major water changes were experienced in the extracellular compartment. The slight changes at the intracellular level indicate that the cell membrane remained intact in spite of the demands of the race.

### 4.3. Muscle-Localized Changes

The muscle group most sensitive to bioelectrical changes induced by the race were the calves, since R/H, Xc/H and Z/H significantly decreased at the middle of the race, indicating a fluid gain in the surrounding area, and reversing the bioelectrical trend during the second half, indicating a fluid loss. Regarding the quadriceps, none of the bioelectrical parameters significantly changed across the race, while the hamstrings significantly increased in PhA in the second half of the race (MID-POST) only. Despite PhA changes only being significant in the hamstrings at MID-POST (4.3 ± 6.1%, *p* = 0.05), a similar non-significant trend was also seen both in the quadriceps (7.7 ± 10.3%, *p* = 0.085) and the calves (6 ± 9.2%, *p* = 0.097), which could imply that the greatest fluid shifts were provoked during the second half of the race, with water loss coming mainly from the extracellular compartment. In Figure 3, the null migration vector of both quadriceps and hamstrings can be observed (Figure 3B,C), while the mean calves’ vector significantly shortened in the first half of the race, and lengthened in the second half, below the baseline values (Figure 3D).

Mascherini et al. [36] used ML-BIVA (analyzing also quadriceps, hamstrings and calves) as a method to assess long-term adaptations in 59 athletes of an elite Italian soccer team after 50 days of training. These players experienced a significant decrease in R (indicating an extracellular local fluid gain) in all three muscles groups studied (3–4.7%). However, only in the calves was there a significant decrease of the Xc (4.1%, *p* = 0.007). PhA remained constant, though. Results suggested that the calves were the most affected muscle group after 50 days of training. As in the present study, it should be analyzed whether the significance of these changes was due to the fact that the properties of the calves are more sensitive than the other muscle groups in the bioelectrical assessments, or because the calf muscles have been the most affected at the cellular level due to the high demands.

However, it is clear is that ML-BIVA is sensitive to some muscle adaptations evoked by the intensity of the race, since the gastrocnemius is one of the most severely affected muscles by fatigue during cycling [37].

### 4.4. Hematological Changes

As shown in Table 2, all the hematological variables were in agreement, reporting an initial gain fluid during the first half of the race, and a posterior dehydration due to the demands of the competition, as reported in the bioelectrical analysis.

The decrease in Hb, and, consequently, in Hct was the expected kinetic due to the hemodilution typical of such heavy efforts, as reported in other Giro d’Italia races [13,38]. According to the aforementioned studies, after the initial decrease in Hb, this parameter stabilizes for a period of time until the race becomes so demanding that reverses the kinetic, increasing both Hb and Hct. High levels of Hb and Hct are an indicator of the dehydration [39] that our cyclist has experienced, especially during the second half of the race, as it can also be confirmed with the vector migration (Figure 3A). Regarding the bioelectrical results, and knowing that R is dependent on lean tissue mass and tissue hydration, Xc is associated with cell size and integrity of the cell membranes, and an increase in EPV in the first phase of the race should be expected as well as a significant decrease at the end [40], according to the results. The osmolality also presents a behavior that indicates the dehydration of the athletes, although this increase occurs more progressively and without correlation with respect to the rest of the bioelectrical parameters, since the significant change was only seen at PRE-POST (PRE-POST: 9.3 ± 10.8%, *p* = 0.011).

The most interesting fact regarding the hematological analysis is that some of these changes are correlated with some WB-BIVA parameter changes. More specifically, the highest correlations are given with the red blood cell parameters and with the EPV. Similarly, Giorgi et al. [41] found some large negative correlation of Z/H and Xc/H changes with EPV changes in some stages of the race, indicating possible fluid shifts. On the other hand, no correlations were found between the changes produced at the muscular level and the hematological changes, so ML-BIVA was shown not to be a good indicator for assessing hydration at a global level, and at a muscle level more research is also needed.

In accordance with this, it is well established that WB-BIVA has shown to be an adequate alternative to detect and rank changes in tissue hydration status, and to be more sensitive than other conventional methods of body composition analysis based on the BM or classical BIA approximation, which cannot detect slight hydration changes.

The study has its five limitations. (1) Sample size was scarce and composed by men only, but that is difficult to amend given that the subjects are world class, and the limited sample also means that the results of this study are considered just preliminary. (2) There is a lack of information of the physical condition of the cyclists prior to race. (3) The lack of information regarding the hydration and nutrition strategies information both prior to and during the race. (4) The lack of length segment measurements to normalize ML-BIVA. (5) The assessments we performed were only in three different moments, but the ideal would have been to monitor the entire competition, as it was composed of different profile stages that could have produced different adaptations.

### 4.5. Practical Applications and Future Perspectives

The 3-week multi-stage cycling races such as the “Giro d’Italia” require constant care and supervision from medical and technical staff to monitor a cyclist’s health and performance in real time. This can only be achieved sing a mobile laboratory to perform the pertinent tests with quick results. For these reasons, simple and reliable tests to assess hydration status, such as bioimpedance monitoring, can be useful to give some orientation and, consequently, to decide the best strategies regarding the health and performance of cyclists. Currently, we are still far from considering BIVA as the unique biomarker of hydration status, but, undoubtedly, its progressive use as a complementary measure to fully validated hematological hydration indicators allows us to parametrize its values and reflect its real possibilities in the near future. The present study could help physical staff to assess the status of their athletes in a precise, accurate, reliable, non-invasive, portable, cheap, and safe way, which could easily be used in real time.

## 5. Conclusions

BIVA appears to be sensitive enough to detect hydration changes induced by a 3-week multi-stage cycling race such as the Giro d’Italia. The bioelectrical kinetics of the whole-body bioimpedance vector coincides and is related with some hematological hydration biomarkers. The preliminary findings encourage BIVA applicability, previously validated in clinical settings, to detect hydration changes induced by multi-stage cycling races. ML-BIVA also appears to be sensitive to local muscle changes by exercise, despite more research being needed in order to better understand the reason for the bioelectrical changes. The calves were the most sensitive to ML-BIVA changes.

## Figures and Tables

**Figure 1 biology-12-00450-f001:**
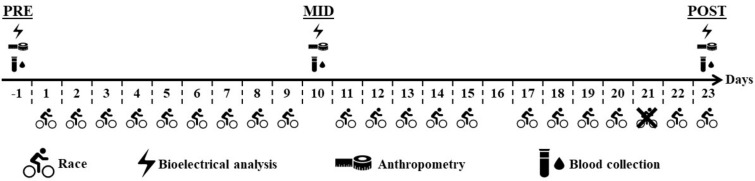
Graphical design of the study. PRE, first assessment one day before the start of the race; MID, second assessment at the middle of the race; POST, last assessment at the end of the race.

**Figure 2 biology-12-00450-f002:**
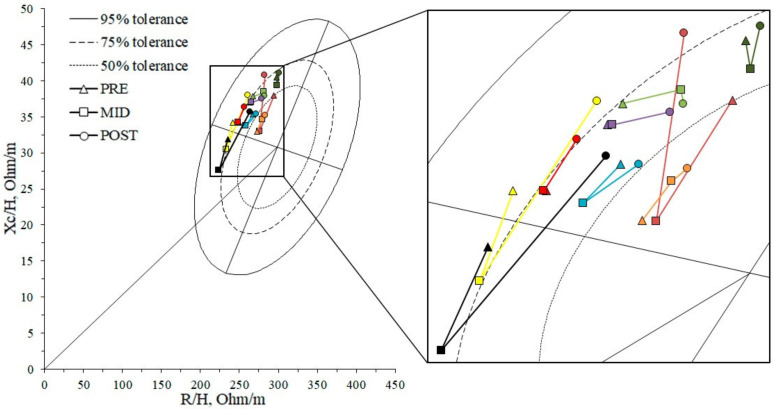
WB-BIVA Z-score point graph. Vector migration subjects over time were plotted with respect to the tolerance ellipse of the male adult Italian reference population [15]. R/H, height-adjusted resistance; Xc/H, height-adjusted reactance.

**Figure 3 biology-12-00450-f003:**
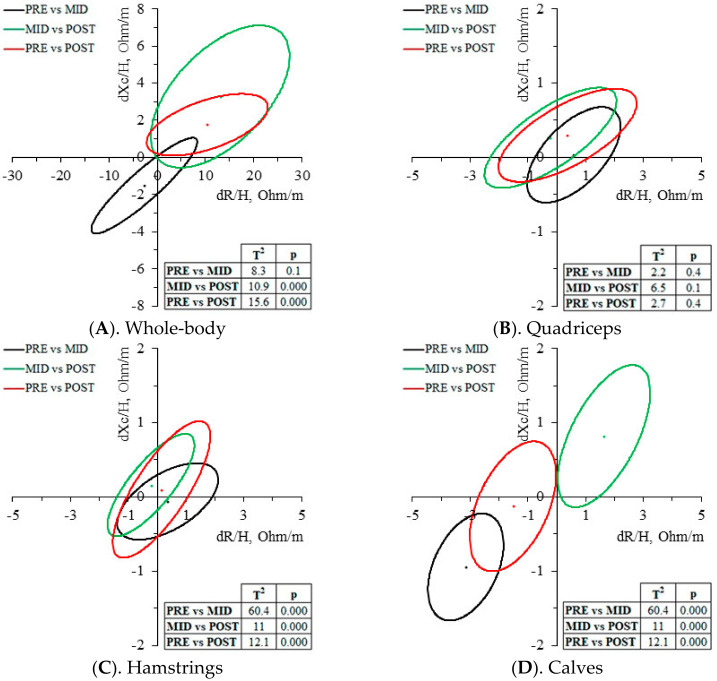
BIVA paired graph. Vector migration across the race. PRE, first assessment at the beginning of the race; MID, second assessment at the middle of the race; POST, last assessment at the end of the race; dR/H, height-adjusted resistance differences; dXc/H, height-adjusted reactance differences. T^2^, Hotelling’s T-squared distribution; p statistical significance, *p* < 0.05.

**Table 1 biology-12-00450-t001:** Main features of the Giro d’Italia 2013 route.

Day	Stage (#)	Length (km)	Accumulated Slope (m)	Profile
−1 *	–	–	–	–
1	1	130.0	1274	F
2	2	17.4	329	Team TT
3	3	222.0	3308	MM
4	4	246.0	2528	MM
5	5	203.0	981	F
6	6	169.0	399	F
7	7	177.0	3664	MM
8	8	54.8	889	Individual TT
9	9	170.0	4786	MM
10 *	Rest	–	–	–
11	10	167.0	4183	HM
12	11	182.0	4235	MM
13	12	134.0	1411	F
14	13	254.0	2390	F
15	14	180.0	2790	HM
16	Rest	–	–	–
17	15	145.0	7095	HM
18	16	238.0	5798	MM
19	17	214.0	1136	F
20	18	20.6	1045	Individual TT
21	19 (Cancelled)	–	–	–
22	20	210.0	3354	HM
23 *	21	206.0	575	F
Total	21	3455.0	52,170	
Mean		164.5	2609	

F, flat; MM, medium mountain; HM, high mountain; TT, time trial; *, measurement days.

**Table 2 biology-12-00450-t002:** Hematological, anthropometric, and bioelectrical changes of elite cyclists during the Giro d’Italia 2013.

				PRE-MID		MID-POST		PRE-POST	
	PRE	MID	POST	∆ (%)	*p*	∆ (%)	*p*	∆ (%)	*p*
Body composition									
BM (kg)	70.5 ± 6.1	70.4 ± 6.2	70.7 ± 5.9	−0.1 ± 1.2	0.574	0.4 ± 1.3	0.400	0.3 ± 1.6	0.499
BMI (kg/m^2^)	21.4 ± 0.8	21.4 ± 0.8	21.4 ± 0.7	−0.1 ± 1.2	0.589	0.4 ± 1.3	0.443	0.3 ± 1.6	0.673
TBW (L)	45.9 ± 3.7	46.1 ± 4.0	45.5 ± 3.3	0.3 ± 1.6	0.398	−1.2 ± 2.1	0.075	−1.0 ± 1.5	0.086
ECW (L)	15.3 ± 1.2	15.3 ± 1.2	15.1 ± 1.0	0.3 ± 1.7	0.683	−1.3 ± 2.2	0.105	−1.1 ± 1.6	0.064
ICW (L)	30.7 ± 2.5	30.8 ± 2.8	30.3 ± 2.2	0.3 ± 1.4	0.458	−1.0 ± 1.9	0.090	−0.8 ± 1.4	0.079
ECW/ICW ratio	0.50 ± 0.01	0.50 ± 0.01	0.50 ± 0.00	−0.1 ± 0.3	1.000	0.3 ± 0.3	0.083	0.2 ± 0.3	0.083
Whole-body BIVA									
R/H (Ω/m)	265.5 ± 21.5	262.8 ± 24.4	275.9 ± 13.9	−1.0 ± 3.7	0.515	5.4 ± 5.9	0.008 *	4.2 ± 4.6	0.038
Xc/H (Ω/m)	35.8 ± 2.8	34.3 ± 3.7	37.5 ± 2.2	−4.3 ± 6.8	0.108	10.5 ± 11.8	0.013 *	5.1 ± 4.6	0.018
Z/H (Ω/m)	267.9 ± 21.6	265.0 ± 24.6	278.4 ± 13.9	−1.1 ± 3.8	0.515	5.5 ± 6.0	0.008 *	4.2 ± 4.6	0.038
PhA (°)	7.7 ± 0.4	7.4 ± 0.4	7.8 ± 0.4	−3.4 ± 4.0	0.048 *	4.7 ± 7.7	0.089	1.0 ± 4.9	0.623
Mean quadriceps BIVA									
R/H (Ω/m)	14.7 ± 1.3	15.3 ± 1.3	15.1 ± 2.4	4.6 ± 10.6	0.236	−1.4 ± 13.6	1.000	2.8 ± 15.9	0.624
Xc/H (Ω/m)	5.1 ± 0.5	5.2 ± 0.5	5.4 ± 0.5	1.3 ± 12.2	0.671	5.8 ± 12.5	0.256	6.4 ± 11.2	0.122
Z/H (Ω/m)	15.6 ± 1.4	16.1 ± 1.3	16.0 ± 2.4	4.2 ± 10.4	0.236	−0.6 ± 13.1	0.722	3.2 ± 14.9	0.553
PhA (°)	19.4 ± 1.6	18.7 ± 1.3	20.1 ± 2.0	−3.1 ± 8.4	0.314	7.7 ± 10.3	0.085	4.2 ± 11.9	0.477
Mean hamstrings BIVA									
R/H (Ω/m)	14.5 ± 2.0	14.9 ± 1.1	14.6 ± 1.6	3.9 ± 13.7	0.463	−1.4 ± 9.4	0.766	2.0 ± 13.1	0.675
Xc/H (Ω/m)	4.9 ± 0.4	4.9 ± 0.5	5.0 ± 0.9	−1.1 ± 9.3	0.608	3.3 ± 13.0	0.440	2.3 ± 17.0	0.866
Z/H (Ω/m)	15.3 ± 2.0	15.6 ± 1.1	15.5 ± 1.7	3.3 ± 12.6	0.574	−0.9 ± 9.6	0.953	2.0 ± 13.2	0.722
PhA (°)	18.9 ± 1.9	18.1 ± 2.0	18.9 ± 2.1	−3.9 ± 9.3	0.262	4.2 ± 6.1	0.050 *	0.0 ± 8.8	0.953
Mean calves BIVA									
R/H (Ω/m)	36.2 ± 3.3	33.1 ± 3.3	34.7 ± 3.4	−8.7 ± 3.2	0.008 *	5.0 ± 4.5	0.017 *	−4.2 ± 3.7	0.018 *
Xc/H (Ω/m)	8.6 ± 1.2	7.6 ± 1.2	8.5 ± 1.2	−10.9 ± 7.3	0.007 *	11.5 ± 12.7	0.028 *	−1.1 ± 9.8	0.357
Z/H (Ω/m)	37.2 ± 3.4	33.9 ± 3.4	35.7 ± 3.6	−8.8 ± 3.3	0.008 *	5.3 ± 4.7	0.013 *	−4.1 ± 3.7	0.017 *
PhA (°)	13.3 ± 1.1	13.0 ± 1.2	13.7 ± 1.0	−2.5 ± 5.8	0.286	6.0 ± 9.2	0.097	3.1 ± 8.7	0.498
Hematological									
Hb (g/dL)	14.1 ± 0.7	13.1 ± 0.9	13.6 ± 0.8	−7.3 ± 3.5	0.008 *	4.1 ± 3.0	0.007 *	−3.6 ± 2.6	0.012 *
Hct (%)	42.0 ± 2.3	39.3 ± 2.3	40.6 ± 2.2	−6.4 ± 2.7	0.008 *	3.6 ± 3.5	0.015 *	−3.1 ± 3.3	0.028 *
EPV (mL)	2663.3 ± 280.2	2785.4 ± 309.4	2730.2 ± 257.9	13.1 ± 6.2	0.011 *	−5.9 ± 4.5	0.050 *	6.2 ± 5.4	0.028 *
POsm (mOsm/L)	285.4 ± 6.2	292.2 ± 16.9	312.2 ± 33.1	2.4 ± 5.5	0.362	7.2 ± 13.0	0.155	9.3 ± 10.8	0.011 *
Na^+^ (mEq/L)	138.7 ± 1.9	141.0 ± 1.4	143.4 ± 6.0	1.7 ± 1.6	0.027 *	1.7 ± 4.0	0.233	3.5 ± 4.7	0.049 *
K^+^ (mEq/L)	5.1 ± 0.2	4.7 ± 0.3	4.8 ± 0.3	−9.3 ± 5.9	0.012 *	3.2 ± 7.3	0.396	−6.7 ± 5.8	0.017 *

BIVA, bioelectrical impedance vector analysis; BM, body mass; BMI, body mass index; ECW, extracellular water; EPV, estimated plasma volume; Hb, hemoglobin concentration; Hct, hematocrit; ICW, intracellular water; K^+^, plasma potassium concentration; MID, second assessment at the middle of the race; Na^+^, plasma sodium concentration; PhA, phase angle; PRE, first assessment one day before the start of the race; POsm, plasma osmolality; POST, last assessment at the end of the race; R/H, height-adjusted resistance; TBW, total body water; Xc/H, height-adjusted reactance; Z/H, height-adjusted impedance; Δ, delta value (%); * *p* statistical significance, *p* < 0.05.

## Data Availability

All data generated analyzed during the current study are available from the corresponding author on reasonable request.

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
