# Peer review of "Bioelectrical, Anthropometric, and Hematological Analysis to Assess Body Fluids and Muscle Changes in Elite Cyclists during the Giro d’Italia"

_biology, 2023, doi:10.3390/biology12030450_

Round 1

Reviewer 1 Report

BIOELECTRICAL, ANTHROPOMETRIC AND HEMATOLOGICAL ANALYSIS TO ASSESS BODY FLUIDS AND MUSCLE CHANGES IN ELITE CYCLISTS DURING THE GIRO D’ITALIA

General Commentary

This article presents a very interesting and pertinent question of research of the investigate characterize, and monitor the body fluid and muscle changes during the Giro d’ Italia in 9 elite cyclists via bioelectrical (whole-boy and muscle-localized), anthropometric and hematological analysis.

Congratulations of the excellent manuscript

However, some questions need to be clarified in order to better understand and apply the results found.

MINOR CONSIDERATION

METHODS

Experimental Design

I suggest the authors insert a first subchapter (Experimental Design) and a figure (with a timeline) in the experimental design of the study, this will make it easier for the reader to understand what was done.

Sample Calculation

I suggest inserting the sample calculation and the sample size.

TABLES

I suggest that authors improve the formatting of tables, using line separation when necessary and placing the entire table on one page. If it is not possible to have the whole table on one page, please separate the table into 1a, 1b continuation....

FIGURES

Figures, in some cases increase the font size of the figures, some were too small the font in relation to the text font.

DISCUSSION

At the beginning of the discussion, I suggest that the authors resume the objective and summarize the results of the present study in text format (first paragraph).

Limitations

I suggest that the authors address possible limitations of the study, such as the sample size.

What is the limitation of the article, having evaluated only 9 athletes, even if they are elite? Also, is there any limitation of including different cycling specialties in the same sample, which one? please add in limitations if needed.

Reviewer 2 Report

I would like to thank you for the opportunity to review this manuscript, which interestingly discusses the findings of a study conducted on professional cyclists during the "Giro d'Italia" competition, to evaluate the changes in BIVA parameters before, during, and after a race.
The study has a scientific sound and despite a limited sample was available, it is not easy to obtain large samples in such specific conditions and competition, therefore I think that it is worth a pilot study.

Maybe, just to be more cautious, I would explain in the conclusions that these are preliminary observations that encourage the use of BIVA [...].

I have only a few minor questions:

- if I am not wrong, the authors did not report the actual fluid intake from the included participants during the race. Was it not possible to collect? Similarly, urinary parameters like USG were not collected. In that case, I suggest discussing it as it seems that most athletes, even at a high competition level, are not well hydrated even before the start of a competition and they continue to drink less than the required amount (10.1123/ijsnem.2015-0188; 10.23736/S0022-4707.16.06722-0), and it might be questionable if this could influence the findings considering the timing between drinking and BIVA observed adjustments.

Thank you again, and congratulations for this interesting work
